# Anxiety and Worries among Pregnant Women during the COVID-19 Pandemic: A Multilevel Analysis

**DOI:** 10.3390/ijerph18136875

**Published:** 2021-06-26

**Authors:** Sara Esteban-Gonzalo, María Caballero-Galilea, Juan Luis González-Pascual, Miguel Álvaro-Navidad, Laura Esteban-Gonzalo

**Affiliations:** 1Psychology Department, Faculty of Biomedicine, Universidad Europea de Madrid, 28670 Madrid, Spain; 2Nursing Department, Faculty of Biomedicine, Universidad Europea de Madrid, 28670 Madrid, Spain; maria.caballero@universidadeuropea.es (M.C.-G.); juanluis.gonzalez2@universidadeuropea.es (J.L.G.-P.); lesteb05@ucm.es (L.E.-G.); 3Obstetrics and Gynecology Department, Faculty of Medicine, Universidad Autónoma de Madrid, 28049 Madrid, Spain; malvaro@fjd.es; 4Department of Obstetrics and Gynecology, Hospital Universitario Fundación Jiménez Díaz, 28040 Madrid, Spain; 5Nursing Department, Faculty of Nursing, Physiotherapy and Podiatry, Universidad Complutense de Madrid, 28040 Madrid, Spain

**Keywords:** anxiety, worries, pregnancy, COVID-19, Spain

## Abstract

*Background***.** Several studies have identified pregnant women as a vulnerable group during the COVID-19 pandemic. The perinatal period has been identified as a stage of great risk for the mental health of pregnant women, due to a large increase in mental pathologies during this period. In this context, the objective of the present study was to assess the associations between socioeconomic and demographic factors, health concerns and health information management, and anxiety level during the COVID-19 pandemic in pregnant Spanish women. *Method***.** The sample of this cross-sectional study was comprised of 353 pregnant women, aged 18 or older and residing in Spain. Data collection was carried out from 1 June to 30 September 2020. Participants were recruited from Quirónsalud University Hospital of Madrid. Multilevel regression models were built to value the associations between demographic factors, health concerns and health information management, and anxiety level during the COVID-19 pandemic among pregnant women. *Results***.** Reduced working hours and income due to the COVID-19 pandemic were related to increased anxiety levels, as was the level of concern about COVID-19 symptoms, potential complications, contagion and consequences for the baby. Worries caused by restrictive measures adopted against COVID-19 and resulting isolation, delivery, postpartum and breastfeeding were also associated with increased anxiety levels. Being a separated or divorced woman and being informed to a greater extent by a midwife were related to lower anxiety levels. An increase in the degree of information obtained about COVID-19 symptoms, complications, contagion and consequences for the baby, restrictive measures and isolation adopted against COVID-19, delivery, postpartum and breastfeeding, were also related to decreased anxiety levels. *Conclusions***.** The most vulnerable future mothers in terms of anxiety levels are those with reduced working hours and income due to the COVID-19 pandemic, those with a higher level of concern and who had access to a lesser degree of information about COVID-19 (symptoms and complications, contagion and consequences on the baby, restrictive measures and isolation, delivery, postpartum and breastfeeding), as well as pregnant women who have obtained information about COVID-19 during pregnancy from TV.

## 1. Introduction

The public health consequences of the COVID-19 pandemic have been devastating [1,2]. The number of deaths from COVID-19 has reached unimaginable numbers. Currently, 3.42 million people have died and 165 million have been infected worldwide to date because of the virus (3.63 million of infected people and 79,568 deaths in Spain). In addition, the health of the uninfected population has also been seriously affected. Specifically, the negative consequences for mental health are still incalculable, with an upward trend in the risk of manifesting a variety of pathologies [3,4,5]. Drastic changes in people’s lifestyles have seriously impacted their way of life [6,7,8]. Social and mobility restrictions, quarantine periods and lockdowns have forced the population into strict isolation conditions [9,10]. In addition to fear of the virus itself, the uncertainty generated by the closure of educational centers, services, stores, and many other businesses has caused a social and economic crisis, seriously affecting people’s mental health and significantly increasing the risk of reporting mental pathologies [11,12,13].

Many studies have yielded important data on how these aspects have influenced the manifestation of psychological distress. For instance, some have emphasized the importance of certain sociodemographic variables and identified groups of greater vulnerability. According to previous studies, more vulnerable populations in terms of mental health during the COVID-19 outbreak were found to be women, younger people, people with basic or medium studies, students and individuals with no remunerated activities, single populations and those co-living with dependent seniors as well as those with a reduced number of children [13,14,15]. Other factors, such as pre-existing physical and mental health conditions and low social support, have been linked to depressive symptoms [16]. Information received, prevention measures, beliefs, concerns, and the population’s knowledge about COVID-19 have also been associated with psychological health [17].

### Pregnant Women during COVID-19 Pandemic

Several studies have identified women as a vulnerable group during the COVID-19 pandemic [13,18,19]. Numerous studies have found higher levels of stress, anxiety, depression, and post-traumatic stress disorder among women, when compared to men [18,20,21]. In addition, an increased risk of family violence during the COVID-19 pandemic has been observed [18]. Other factors of social vulnerability must also be taken into account. Lack of social support and isolation have contributed negatively to the mental health status of women [19]. In many cases, household and childcare tasks have been delegated to women in a more pronounced way, increasing pre-existing levels of stress and anxiety [20,21,22]. 

Pregnancy and postpartum mental illnesses include depression, anxiety disorders, and postpartum psychosis, which usually manifests as bipolar disorder [23]. Perinatal depression and anxiety are common [24]. Long-term psychiatric complications of pregnancy and postpartum mental illness include unipolar major depression, bipolar depression and obsessive-compulsive symptoms, among others [25]. Psychological, social, behavioral, biological and environmental forces shape mental disorders during the perinatal period, but also provide a window into a woman’s long-term health [26,27]. Sleep deprivation, hormonal changes and the pressure of caring for a new infant may enhance the development of pregnancy and postpartum mental illness and unmask a psychological vulnerability leading to psychiatric disease later in life [25].

Several studies have investigated women’s mental health during the COVID-19 pandemic [28,29,30,31,32]. Given the importance of stressors on the mental health of women in the perinatal period, this group is presented as one of high risk for developing a variety of psychopathologies [19]. Stressors during the COVID-19 period for pregnant women have been numerous. For example, restrictions in hospitals have led many women to make the decision to give birth at home, exposing them to risks such as complications in childbirth without adequate care [29]. Although transmission of SARS-COV-2 through breast milk is not common, some women have decided not to breastfeed their babies to reduce the risk of infection [33,34]. The virus containment measures enacted by governments have increased feelings of isolation, confusion and anxiety in an already vulnerable population [29,35]. Economic and financial uncertainties may have worsened feelings of uncertainty, thus increasing pregnant women’s vulnerability even more [29].

According to recent studies carried out in Europe, the prevalence of depressive symptoms in pregnancy and postpartum was explicitly higher compared to data obtained prior to the pandemic [36]. Longitudinal studies in Spain have observed a gradual increase in psychopathological indicators and a decrease in positive affect among pregnant women during the strict lockdown period [30]. Studies conducted in China have shown an increase in self-harm thoughts, and in depressive symptoms positively associated with the number of newly-confirmed cases of coronavirus, suspected infections, and deaths per day [37,38]. Some associated risk factors of mental illness among pregnant women were isolation and loneliness [39], stress and loss of income and violence at home, among others [40,41,42].

The results regarding the sociodemographic correlates that could exacerbate psychological distress in pregnant women are not conclusive. For instance, in studies carried out in Argentina and China, no significant differences in the risk of psychological distress seemed to have been observed in relation to age, socioeconomic status, educational level, number of children, and dwelling size [30,43]. In contrast, other studies carried out in China did find a relationship between these same variables and the mental health of pregnant women, emphasizing the protective factors of age, education and a well-paid occupation for their mental health [44]. Other studies carried out in the United States have found that income loss due to the pandemic and being a woman of color were associated with greater levels of stress [45]. In addition, the same authors found that access to outdoor space and an older age were protective factors against stress [45].

Health concerns should not be underestimated either. Health concerns for oneself or one’s infant, such as accessibility to health services, availability of transportation, or the absence of tests to monitor infant health, seem to have taken on an inestimable influence on women’s mental health concerns [46,47,48]. The number of children born, complication during pregnancy, availability of prenatal care services, and use of social media for obtaining health information were factors associated to pregnant women’s mental health [44]. In addition, the way in which the COVID-19 pandemic is experienced should also be highlighted. Loneliness and feelings of vulnerability such as fear of infection, uncontrollable stress and perceiving unpredictable environments—including economic and health stressors—may also contribute to poorer women’s psychological health [49].

Therefore, taking into account the evidence provided by the aforementioned studies, indicating significant vulnerability of pregnant women in terms of mental health, our research question is aimed at detecting the factors that may influence anxiety levels of pregnant women living in Spain during the COVID-19 pandemic. Thus, the objective of the present study was to identify those socioeconomic, demographic, health-related and information management factors and concerns that may influence the anxiety level of pregnant women during the COVID-19 pandemic.

## 2. Method

### 2.1. Study Design and Participants

This is a cross-sectional study designed to value the associations between social and demographic factors, health concerns and health information management, and anxiety level of pregnant women in Spain during the COVID-19 pandemic.

Participants were recruited from the Quirónsalud University Hospital of Madrid. This hospital leads the initiative of offering pregnant women the possibility of subscribing to a monthly newsletter that provides them health information about each stage of their pregnancy. Other hospitals have joined this initiative. Among them, the Jiménez Díaz Foundation University Hospital and San José Quirónsalud Hospital also decided to collaborate in the present study. After obtaining permission from each institution, complete information on the study was provided in the newsletter. Women who accepted to participate in the study had access to the anonymous questionnaire and were recruited through a consecutive convenience sample.

Inclusion criteria were to be pregnant, to be 18 years or older, to reside in Spain, to be able to fill out the Spanish questionnaire and to provide written informed consent. Data were collected between 1 June and 30 September 2020, a period during which Spain, among many other countries, was fighting against the COVID-19 pandemic.

A total of 453 women completed the questionnaire, of which, 45 women had already given birth to their babies and were excluded. Of the remaining 408 women, 353 presented valid data about their anxiety level and were included in the analysis. The sample size was calculated for a linear regression using the G-Power tool, considering an Alpha error of 0.05 and a 0.95 statistical power, resulting in a minimum sample size of 262 women. Post hoc statistical power calculations were also carried out, for an Alpha error of 0.05 and according to the effect size range obtained in the models (considering the three predictors used), which showed a statistical power higher than 0.95 in all cases.

### 2.2. Measurement Instruments

#### 2.2.1. Anxiety Level

The dependent variable, anxiety level, was assessed using the State Trait Anxiety Inventory (STAI). This tool was employed for three different reasons. First, it has been commonly used in relevant studies, widely cited in the literature and adapted for use in a large number of countries [50]. Second, it is a well-documented scale capable of diagnosing anxiety in clinical settings and can be used to detect distress. Third, the Spanish adaptation has shown adequate psychometric properties, with a Cronbach’s alpha reliability of 0.90 for Trait and 0.94 for State Anxiety, and has also been sensitive to increased environmental stimuli that produce stress [51]. In the present study, only STAI state (STAI-S) was used due to limitations in the length of the questionnaire. It is composed of 20 items with a 4-point scale (0–3 points) and a final score between 0 and 60. A higher score corresponds to increased anxiety level [52].

#### 2.2.2. Socioeconomic and Demographic Factors

Several socioeconomic and demographic conditions have been considered independent variables: − Number of previous children were reported as a number by the participants.− Country of origin was collected using a single question: what is your country of origin? (Spain/other).− Area of residence was determined from the city code referred by the participants. Each area of residence was dichotomized in rural or urban area of residence according to the National Institute of Statistics definition (urban areas those with more than 10,000 inhabitants and rural areas with less) [53].− Level of education was reported as basic level of studies (primary and secondary school), medium level (baccalaureate and technical education) and high level (university studies).− Marital status was reported by the participants as: married, single, unmarried partner, separated/divorced and widowed. No widowed women were identified; thus, this category will not be included in the tables and results section.− Employment status during the COVID-19 pandemic was referred by the participants as: self-employment, employment, unemployment, homemaker, or student. No students were detected.− To have lost one’s job/have reduced working hours/have reduced income due to the COVID-19 pandemic was reported as yes or no through the following questions: Have you lost your job/Have your working hours been reduced/Has your salary been reduce due to the COVID-19 pandemic?

#### 2.2.3. Pregnancy-Related Factors

The following factors associated with the gestation period have been taken into account as independent variables:− Trimester of pregnancy was reported as first, second or third.− Week of pregnancy was referred as a number by the participants.− Primiparous women were considered those who reported no previous children.

#### 2.2.4. Health-Related Concerns

The participants were asked to rate from 1 to 5 their level of concern about: COVID-19 symptoms and complications of the sickness, contagion and consequences on the baby, restrictive measures and isolation adopted against COVID-19, delivery, and postpartum and breastfeeding.

#### 2.2.5. Health Information-Related Factors

On the one hand, participants were asked where they obtained information with the following question: Where did you look for or obtain information about COVID 19 and pregnancy? Please point out those answers that are most relevant. Possible answers were: health professionals they were treated by, hospital obstetric newsletter, hospital obstetric web page, official sources of information (for example information provided by the Spanish Ministry of Health or the government), internet, social networks, family and friends and TV. Each woman was able to choose as many answers as she considered appropriate. Based on this information, eight different variables were constructed, one for each response category. All of them were dichotomous (yes/no), referring to whether each participant obtained or sought information from each potential source.

On the other hand, each participant was asked to rate from 1 to 5 the level of information they obtained about: COVID-19 symptoms and complications of the sickness, contagion and consequences on the baby, restrictive measures adopted against COVID-19 and resulting isolation, delivery, and postpartum and breastfeeding. As a result, five separate variables were considered as to the degree of information available to the participants on each item assessed. 

Health professionals who provide information were also reported: gynecologist, midwife, nurse or other (administrative staff, pharmaceutical, etc). In this regard, four dichotomous variables (yes/no) were available for analysis, based on whether each participant obtained information from each of the specified professionals. 

Finally, the level of satisfaction with the information provided by health workers, and with the empathy shown while informing, was rated from 1 to 5 by participants. Therefore, two variables were considered. The first variable refers to the degree of satisfaction with the information provided by the professional(s) previously specified, regardless of the professional from whom the information was obtained. The second was based on the degree of empathy shown by the professional(s) who provided the information. 

All these variables have been considered independent variables. 

### 2.3. Co-Variates

Additionally, age and medical COVID-19 diagnosis were considered co-variates in the analyses performed. Medical COVID-19 diagnosis (yes/no) was assessed asking participants if they had a positive COVID-19 test. Age was referred as a number by each participant. Both variables were included as co-variates for the analysis, given their relevance and potential influence on anxiety level.

### 2.4. Ethical Procedures

The protocol for the present study obtained approval from the Ethics Committee of the Jiménez Díaz Foundation Hospital (E0070-20_HUQM). All institutions and participants were informed of the purpose and intent of the study and provided written consent. Similarly, anonymity of each of the participants was ensured.

### 2.5. Data Analyses

All statistical analysis was conducted using the Statistical Package for the Social Sciences software version 21.0 (SPSS. Inc., Chicago, IL, USA) and STATA/SE 14.1 software (Stata Corp LP, College Station, TX, USA).

Descriptive statistics (mean values and standard deviations or numbers and percentages) were calculated to describe participant characteristics. Differences between categorical variables and anxiety score were assessed using the Mann–Whitney U test for dichotomous variables and the Kruskal-Wallis test for variables with more than two categories. The Spearman correlation test was employed to value associations between quantitative variables and STAI-S scores after assessing the distribution of each variable using the Kolmogorov-Smirnov test (all, *p* < 0.001).

Multilevel linear regression was used to test the association between social and demographic factors, health concerns and health information-related variables, and STAI-S score. Non-parametric variables were transformed to address normality. All models included a random intercept for the hospital where the participant was recruited. First, unadjusted models were constructed to assess the influence of each of the independent variables studied on the dependent variable (anxiety levels). Secondly, it was considered necessary to adjust the models for two variables that could influence the anxiety levels of the participants, in order to give greater validity to the results obtained in this study. The first of these was age. The second was having a medical diagnosis of COVID-19, although very few participants were included in this variable. These variables were included one by one in the models constructed. Since neither of these two variables represented a significant variation in the results, a single model adjusted for both variables was constructed, and only adjusted models will be described in the results section.

## 3. Results

Mean and standard deviation (SD) values of the STAI-S, as well as characteristics of the participants, are presented in Table 1. Pregnant women showed a mean score of 24.6 (10.4 SD). Of the women examined, 2.5% had a medical COVID-19 diagnosis, a status that was not associated to STAI-S score.

### 3.1. Socioeconomic and Demographic Factors

The STAI-S score was higher in women who have reduced working hours (27.8+/−10.9, *p* = 0.005) and income (26.4+/−10.8, *p* = 0.025) due to the COVID-19 pandemic, and lower in separated or divorced women (16.6+/−6.6, *p* = 0.027),

### 3.2. Health-Related Concerns

The STAI-S score was higher in women with an increasing level of concern about COVID-19 symptoms and complications from the sickness (*p* < 0.001, *r* = 0.28), contagion and consequences on the baby (*p* = 0.001, *r* = 0.18), restrictive measures and isolation adopted against COVID-19 (*p* = 0.001, *r* = 0.18), delivery (*p* < 0.001, *r* = 0.24), and postpartum and breastfeeding (*p* < 0.001, *r* = 0.22).

### 3.3. Health Information-Related Factors

The STAI-S score was lower in women with a greater degree of information about COVID-19 symptoms and complications of the sickness (*p* = 0.001, *r* = −0.16), contagion and consequences for the baby (*p* = 0.001, *r* = −0.17), restrictive measures and isolation adopted against COVID-19 (*p* = 0.006, *r* = −0.14), delivery (*p* = 0.007, *r* = −0.14), and postpartum and breastfeeding (*p* < 0.001, *r* = −0.18). Women with a higher satisfaction level with the information provided by health professionals (*p* < 0.001, *r* = −0.18) and with the empathy shown by health professionals when informing (*p* < 0.001, *r* = −0.23) also showed a lower STAI-S score. However, the STAI-S score was higher in women who reported obtaining information about COVID-19 and pregnancy from TV (26.8+/−9.8, *p* = 0.014).

Multilevel linear regression models for the STAI-S score are shown in Table 2. As previously mentioned, since no relevant differences were detected between the unadjusted model and the adjusted model, only the results associated with the adjusted model will be shown in this section.

### 3.4. Socioeconomic and Demographic Factors

Reduced working hours (β = 4.07, 1.35 (SE), *p* = 0.003) and reduced income (β = 2.66, 1.19 (SE), *p* = 0.025) due to the COVID-19 pandemic was related to a higher STAI-S score. However, being a separated or divorced woman (β = −8.33, 2.79 (SE), *p* = 0.003) was related to a lower STAI-S score.

### 3.5. Health-Related Concerns

A one-unit increase in the level of concern (1%) about COVID-19 symptoms and complications of the sickness (β = 2.70, 0.49 (SE), *p* < 0.001), contagion and consequences for the baby (β = 1.87, 0.59 (SE), *p* = 0.002), restrictive measures and isolation adopted against COVID-19 (β = 1.66, 0.46 (SE), *p* < 0.001), delivery (β = 2.08 0.59 (SE), *p* < 0.001) and postpartum and breastfeeding (β = 1.81, 0.49 (SE), *p* < 0.001) was associated with a higher STAI-S score.

### 3.6. Health Information-Related Factors

Obtaining information from TV (β = 2.92, 1.30 (SE), *p* = 0.025) was related to a higher STAI-S score. However, being informed to a greater extent by a midwife (β = −3.05, 1.36 (SE), *p* = 0.025) was related to a lower STAI-S score. A one-unit increase in the degree of information obtained about COVID-19 symptoms and complications of the sickness (β = −1.41, 0.42 (SE), *p* = 0.001), contagion and consequences on the baby (β = −1.37, 0.42 (SE), *p* = 0.001), restrictive measures and isolation adopted against COVID-19 (β = −1.12, 0.40 (SE), *p* < 0.001), delivery (β = −1.16, 0.40 (SE), *p* = 0.004) and postpartum and breastfeeding (β = −1.68, 0.42 (SE), *p* < 0.001) was related to a lower STAI-S score. Finally, a one-unit increase in the level of satisfaction with the information provided by the health professional (β = −1.41, 0.45 (SE), *p* = 0.002) and the empathy shown by the professional when informing (β = −2.02, 0.43 (SE), *p* < 0.001) was associated to a lower STAI-S score.

## 4. Discussion

As previously mentioned, the objective of the present study was to identify those socioeconomic, demographic, health-related and information management factors and concerns that may influence the anxiety level of pregnant women during the COVID-19 pandemic. The results obtained in the study provide important information about those factors that increase anxiety and those that could help to mitigate it. According to our results, reduced working hours and reduced income due to the COVID-19 pandemic were related to higher anxiety levels. These results are congruent with other studies carried out previously, in which the vulnerability of this group of women has been evidenced [32,54]. While economic and employment uncertainty has proven to be a risk factor for mental health in the general population [13,55,56,57,58], it may be even more so in this group. Specifically, it could be hypothesized that, while the stress and pressure associated with new economic responsibilities is often a risk factor for the mental health of pregnant women under normal circumstances [59,60,61], one could expect it to be to be even more so in conditions of economic uncertainty and instability such as those caused by the COVID-19 outbreak. Given the peculiarity of the COVID-19 pandemic, the absence of similar baseline situations and the abruptness of the social changes made, the responsibility of bringing a new life into the world may be increased, in a context in which one’s working hours and salary have been affected. After all, it is known that fear of job loss has been a recurrent concern in pregnant women, significantly increasing levels of stress and anxiety [46].

According to this study’s results, the level of concern about COVID-19 symptoms and complications of the sickness, contagion and consequences for the baby, restrictive measures and isolation adopted against COVID-19, delivery, postpartum and breastfeeding were also associated with higher anxiety levels. While early medical advice was skeptical about the risk of COVID-19 transmission between new mothers and their newborns, subsequent studies have shown that, with the necessary precautions, breastfeeding infants is safe [62,63]. However, several studies have shown that, despite evidence that breast feeding is safe, pregnant and new mothers continue to worry about risk of contagion and infection of the baby to the extent that many women have decided not to breastfeed their children [33,34]. While this study does not provide data on women who have decided to forgo breastfeeding, it is known that the fear of breastfeeding has significantly increased anxiety levels in pregnant women. Uncertainty, fear, and the amount of contradictory information on the subject may have precipitated these states of anxiety among pregnant women [50,64,65]. In addition, face-to-face professional support for breastfeeding was reduced or cancelled in many countries, which may have contributed to increased anxiety levels regarding breastfeeding. Concretely, it has been found that mothers with lower educational levels, more challenging living circumstances and from minority ethnic groups were more likely to be affected by the situation and stop breastfeeding [66].

The absence or restriction of professional support in the case of breastfeeding can be extended to pregnancy, childbirth and postpartum [47,64,67,68]. For instance, other studies have found that discontinuing face-to-face prenatal visits and making new plans to avoid delivering in a hospital may have increased anxiety during the pandemic [46,47]. In addition, lack of ultrasound results and other regular health exams may have also negatively contributed to women’s perceived control, as has been observed in studies prior to the pandemic [48]. Similarly, other authors have hypothesized that a greater locus of control among pregnant women (for instance, by planning childbirth at home) may have helped to reduce anxiety [46].

Finally, concern due to the extensive restrictive measures and social isolation mandated by governments should be highlighted as a risk factor associated with anxiety in pregnant women. Access to public places and transportation has also been a major concern for pregnant women, as reported in other studies [47]. If isolation measures have been shown to damage the mental health of the general population, the same phenomenon has logically been observed in pregnant women [38,65,69,70]. Loneliness and rumination resulting from the lockdown may have been exacerbated by fear of not being able to access health services if the need arose [46,47], as well as by a rise in other threatening factors such as deterioration of relationships with other co-living family members and increased gender-based violence, factors that have been documented by other authors [71−73].

In terms of protective factors for anxiety, according to our results, being a separated or divorced woman and being informed to a greater extent by a midwife were related to lower anxiety levels. An increase in the degree of information obtained about COVID-19 symptoms and complications of the sickness, contagion and consequences on the baby, restrictive measures and isolation adopted against COVID-19, delivery, postpartum and breastfeeding were related to decreased anxiety levels. An increase in the satisfaction level with information provided by the health professional and the empathy shown by the professional, while still informing, were also associated to reduced anxiety scores. These results are in line with previous studies emphasizing the importance of information received during COVID-19 for people’s mental health [15,74,75]. According to these studies, receiving sufficient information regarding the virus was a protective factor in the appearance of symptoms of depression, anxiety and post-traumatic stress disorders [15]. However, the use of the Internet (unofficial web-based media) as a source of information during the COVID-19 was significantly associated with poorer psychological well-being and mental health [71,76]. Therefore, the quality of the information received is a key aspect for determining its impact on mental health [71].

Satisfaction with the information received could also play an important role [77]. An empathetic relationship with healthcare professionals has been proven to benefit patients in numerous research studies [72,78]. However, the COVID-19 pandemic has once again brought these findings to the forefront [73,74]. Empathetic and quality communication with the health professional increases satisfaction and adherence to treatment, and improves objective and subjective outcomes among patients [72], especially in such a difficult context as COVID-19 [75].

Finally, it is worth noting that being single or divorced was found to be a protective factor for anxiety. Although loneliness has been identified as a risk factor for mental health during the COVID-19 outbreak [79], other studies point out that marital conflict and gender-based violence have threatened women’s well-being [65,80], which may help contextualize the findings.

The study presents some limitations that must be taken into account when interpreting the results. Although all questionnaires were carefully chosen and all are valid and reliable, the variables are self-reported, which could bias the inherent quality of the data. In addition, the absence of specific measures to identify and assess mental health during COVID-19 has been a limitation, and researchers have had to rely on already validated, but perhaps not as specific, measures. Unmeasured covariates may have resulted in residual confounding. The cross-sectional nature of the data cannot infer causation; we can only report associations between mental health indicators and social, demographic and economic factors. The sample considered in this study is not representative of the general Spanish population. Thus, the results of the present study cannot be generalizable to the general population. Finally, future longitudinal studies should be carried out to extend the cross-sectional perspective examined in this study.

## 5. Conclusions

In conclusion, the most vulnerable future mothers in terms of anxiety levels are those with reduced working hours and income due to the COVID-19 pandemic, those with a higher level of concern and who had access to a lesser degree of information about COVID-19 (symptoms and complications, contagion and consequences on the baby, restrictive measures and isolation, delivery, postpartum and breastfeeding), as well as pregnant women who have obtained information about COVID-19 during pregnancy from TV. By contrast, being a separated or divorced woman, being informed in a satisfactory manner by an empathetic professional, especially by a mid-wife, was related to more favorable anxiety levels. 

Since maternal mental health problems are associated with short-term and long-term risks for the mothers’ health and their children’s overall development, these results must be seriously considered. As practical implications of these findings, competent health systems should include this information when designing procedures to deal with current and future pandemics. Health authorities, counselors and obstetric care providers should take into account the relevance of the health information provided on the mental health of an already vulnerable group in a pandemic context. Therefore, appropriate protocols should be implemented in order to ensure that adequately trained and empathetic professionals provide sufficient quality information, thus preventing high levels of anxiety in pregnant women.

## Figures and Tables

**Table 1 ijerph-18-06875-t001:** Characteristics of the participants examined and bivariate analyses between STAI-S score and socioeconomic and demographic factors, pregnancy-related factors, health-related concerns and health information-related factors.

n	353	STAI-S Score	p ^a^
STAI-S score (0–60) [mean (SD)]	24.6 (10.4)		
Positive COVID-19 test (%)			0.563 ^#^
Yes	2.5	25.6 (6.8)	
No	97.5	24.5 (10.4)	
Age [mean (SD)]	35.9 (7.0)		0.654 (−0.02) ^x^
Socioeconomic and demographic variables			
Country of origin (%)			0.780 ^#^
Spain	90.7	24.6 (10.4)	
Other	9.3	24.4 (10.0)	
Area of residence (%)			0.099 ^#^
Urban area	94.0	24.8 (10.4)	
Rural area	6.0	20.9 (9.6)	
Level of education (%)			0.129 *
Basic level of studies	3.0	26.1 (7.0)	
Medium level of studies	13.0	27.0 (10.5)	
High level of studies	84.0	24.1 (10.4)	
Marital status			**0.027 ***
Married	66.6	25.1 (10.3)	
Single	22.1	24.6 (10.0)	
Unmarried partner	7.4	23.3 (11.7)	
Separated/divorced	3.9	16.6 (6.6)	
Employment status (%)			0.771 *
Self-employment	9.6	24.4 (9.8)	
Employment	83.0	24.4 (10.5)	
Unemployment	5.1	25.5 (9.6)	
Homemaker	2.3	27.8 (10.3)	
To have lost one’s job ^b^ (%)			0.156 ^#^
Yes	5.1	28.3 (10.9)	
No	94.9	24.3 (10.3)	
Reduced working hours ^b^ (%)			**0.005 ** ^#^
Yes	20.4	27.8 (10.9)	
No	79.6	23.7 (10.1)	
Reduced income ^b^ (%)			**0.025 ** ^**#**^
Yes	30.9	26.4 (10.8)	
No	69.1	23.7 (10.1)	
Number of previous children [mean (SD)]	0.3 (0.5)		0.109 (0.08) ^x^
Pregnancy-related factors			
Primiparous women (%)			0.119 ^#^
Yes	23.2	25.9 (10.2)	
No	76.8	24.1 (10.4)	
Trimester of pregnancy (%)			0.716 *
First	19.2	25.1 (10.6)	
Second	38.4	24.1 (10.8)	
Third	42.4	24.7 (9.9)	
Week of pregnancy [mean (SD)]	24.9 (9.3)		0.462 (−0.03) ^x^
Health-related concerns			
Level of concern about: (1–5) [mean (SD)]			
COVID-19 symptoms and complications	4.1 (1.0)		**<0.001 (0.28) ** ^**x**^
Contagion and consequences on the baby	4.5 (0.9)		**0.001 (0.18) ** ^**x**^
Restrictive measures and isolation	3.3 (1.1)		**0.001 (0.18) ** ^**x**^
Delivery	4.2 (1.0)		**<0.001 (0.24) ** ^**x**^
Postpartum and breastfeeding	4.0 (1.0)		**<0.001 (0.22) ** ^**x**^
Health information-related factors			
Degree of information you have about: (1–5) [mean (SD)]			
COVID-19 symptoms and complications	2.6 (1.2)		**0.001 (−0.16) ** ^**x**^
Contagion and consequences on the baby	2.5 (1.3)		**0.001 (−0.17) ** ^**x**^
Restrictive measures and isolation	3.3 (1.3)		**0.006 (−0.14) ** ^**x**^
Delivery	2.4 (1.3)		**0.007 (−0.14) ** ^**x**^
Postpartum and breastfeeding	2.2 (1.3)		**<0.001 (−0.18) ** ^**x**^
Have obtained information about COVID-19 during pregnancy from: (%)			
Health professionals	36.0	24.9 (9.7)	0.615
Hospital obstetric newsletter	19.5	24.4 (10.4)	0.778 ^#^
Hospital obstetric web page	15.6	23.2 (9.5)	0.409 ^#^
Official sources of health information	25.2	23.6 (10.1)	0.524 ^#^
Internet	72.8	25.0 (10.2)	0.113 ^#^
Social networks	31.4	26.0 (9.9)	0.064 ^#^
Family and friends	19.5	25.1 (10.2)	0.582 ^#^
Tv			**0.014 ** ^#^
Yes	23.5	26.8 (9.8)	
No	76.5	23.8 (10.4)	
Health professional who has provided the most information: (%)			0.204 *
Gynecologist	45.9	24.7 (10.4)	
Midwife	20.1	22.1 (9.5)	
Nurse	4.0	23.7 (9.1)	
Other	13.0	26.2 (9.9)	
Satisfaction with information provided by health professionals (1–5) [mean (SD)]	2.9 (1.2)		**<0.001 (−0.18) ** ^**x**^
Satisfaction with empathy showed by professionals when informed (1–5) [mean (SD)]	3.6 (1.2)		**<0.001 (−0.23) ** ^**x**^

*p ^a^* value for comparing variables studied and STAI-S score. ^x^ Spearman correlation test, *p* (correlation coefficient). # U-Mann-Withney test. * Kruskal-Wallis test. ^b^ Due COVID-19. Bold: *p* ≤ 0.05

**Table 2 ijerph-18-06875-t002:** Multilevel linear regression models for STAI-S score (*n* = 353).

Title	Unadjusted Model				Adjusted Model		
	*n*	β (SE)	95% CI	*p*	β (SE)	95% CI	*p*
Socioeconomic and demographic variables							
Country of origin	353						
Other than Spain		−0.14 (1.89)	−3.86–3.57	0.940	−0.02 (1.09)	−3.99–3.47	0.892
Area of residence	352						
Rural area		−3.86 (2.33)	−8.42–0.70	0.098	−3.82 (3.33)	−8.38–0.74	0.101
Level of education	351						
Basic level of studies		1.63 (3.04)	−4.33–7.61	0.591	1.55 (3.05)	−4.42–7.53	0.610
Medium level of studies		2.82 (1.63)	−0.37–6.03	0.083	2.83 (1.63)	−0.37–6.04	0.084
High level of studies		−2.97 (1.50)	−5.92–0.01	0.049	−2.96 (1.51)	−5.93–0.00	0.050
Marital status	353						
Married		1.78 (1.16)	−0.50–4.06	0.127	1.70 (1.17)	−0.59–4.00	0.146
Single		0.05 (1.33)	−2.55–2.66	0.966	0.18 (1.34)	−2.44–2.81	0.889
Unmarried partner		−1.33 (2.11)	−5.47–2.80	0.527	−1.32 (2.11)	−5.46–2.81	0.531
Separated/divorced		−8.26 (2.79)	−13.74–2.78	**0.003**	−8.33 (2.79)	−13.82–2.84	**0.003**
Employment status	353						
Self-employment		−0.19 (1.87)	−3.86–3.48	0.919	−0.28 (1.87)	−3.96–3.39	0.880
Employment		−0.74 (1.47)	−3.62–2.13	0.613	−0.65 (1.48)	−3.55–2.25	0.661
Unemployment		0.96 (2.51)	−3.95–5.88	0.701	0.92 (2.51)	−3.99–5.85	0.711
Homemaker		3.36 (3.70)	−3.89–10.63	0.364	3.31 (3.79)	−4.12–10.75	0.382
To have lost one’s job ^a^	353	4.00 (2.50)	−0.89–8.91	0.109	3.95 (2.50)	−0.95–8.86	0.114
Reduced working hours ^a^	353	4.09 (1.35)	1.44–6.75	**0.002**	4.07 (1.35)	1.42–6.72	**0.003**
Reduced income ^a^	353	2.72 (1.18)	0.39–5.05	**0.022**	2.66 (1.19)	0.32–5.00	**0.025**
Number of previous children	353	1.30 (1.00)	−0.65–3.27	0.191	1.42 (1.04)	−0.62–6.47	0.174
Pregnancy-related factors							
Primiparous women	353	1.82 (1.30)	−0.72–4.38	0.161	1.93 (1.30)	−0.62.4.50	0.139
Trimester of pregnancy	353						
First		0.71 (1.40)	−2.02–3.46	0.609	0.67 (1.40)	−2.06–3.42	0.628
Second		−0.74 (1.13)	−2.96–1.48	0.514	−0.74 (1.14)	−2.98–1.50	0.518
Third		0.26 (1.11)	−1.92–2.45	0.816	0.27 (1.12)	−1.93–2.48	0.806
Week of pregnancy	353	−0.04 (0.05)	−0.15–0.07	0.470	−0.04 (0.05)	−0.15–0.07	0.471
Health-related concerns							
Level of concern about: (1–5)	353						
COVID-19 symptoms and complications		2.68 (0.49)	1.70–3.66	**<0.001**	2.70 (0.49)	1.72–3.67	**<0.001**
Contagion and consequences on the baby		1.85 (0.59)	0.70–3.01	**0.002**	1.87–0.59	0.71–3.03	**0.002**
Restrictive measures and isolation		1.66 (0.45)	0.76–2.56	**<0.001**	1.66 (0.46)	0.76–2.57	**<0.001**
Delivery		2.09 (0.50)	1.09–3.08	**<0.001**	2.08 (0.59)	1.08–3.08	**<0.001**
Postpartum and breastfeeding		1.79 (0.49)	0.82–2.77	**<0.001**	1.81 (0.49)	0.83–2.78	**<0.001**
Health information-related factors							
Degree of information you have about: (1–5)	353						
COVID-19 symptoms and complications		−1.39 (0.42)	−2.23–0.56	**0.001**	−1.41 (0.42)	−2.24–0.58	**0.001**
Contagion and consequences on the baby		−1.35 (0.42)	−2.19–0.52	**0.001**	−1.37 (0.42)	−2.21–0.54	**0.001**
Restrictive measures and isolation		−1.13 (0.40)	−1.93–0.33	**0.005**	−1.12 (0.40)	−1.92–0.32	**0.001**
Delivery		−1.15 (0.40)	−1.94–0.36	**0.004**	−1.16 (0.40)	−1.95–0.37	**0.004**
Postpartum and breastfeeding		−1.66 (0.42)	−2.49–0.83	**<0.001**	−1.68 (0.42)	−2.51–0.85	**<0.001**
Have obtained information about COVID-19 during pregnancy from:	353						
Health professionals		0.57 (1.15)	−1.67–2.83	0.616	0.62 (1.15)	−1.63–2.88	0.587
Hospital obstetric newsletter		−0.18 (1.39)	−2.91–2.54	0.894	−0.17 (1.39)	−2.90–2.55	0.899
Hospital obstetric web page		−1.55 (1.52)	−4.53–1.42	0.307	−1.55 (1.52)	−4.54–1.43	0.308
Official sources of health information		−1.20 (1.27)	−3.69–1.28	0.345	−1.24–1.27	−3.74–1.24	0.327
Internet		1.84 (1.23)	−0.57–4.27	0.136	1.77 (1.24)	−0.65–4.21	0.152
Social networks		2.17 (1.18)	−0.14–4.49	0.067	2.09 (1.19)	−0.24–4.43	0.079
Family and friends		0.73 (1.39)	−1.99–3.46	0.598	0.68 (1.39)	−2.04–3.41	0.622
Tv		2.98 (1.29)	0.45–5.52	**0.021**	2.92 (1.30)	0.36–5.48	**0.025**
Health professional who has provided the most information:	353						
Gynecologist		0.38 (1.10)	−1.79–2.55	0.731	0.34 (1.10)	−1.82–2.51	0.756
Midwife		−3.05 (1.36)	−5.74–0.37	**0.025**	−3.05 (1.36)	−5.73–0.37	**0.025**
Nurse		−0.90 (2.83)	−6.45–4.64	0.749	−1.10 (2.84)	−6.68–4.47	0.698
Other		1.92 (1.63)	−1.28–5.13	0.239	2.06 (1.64)	−1.16–5.29	0.210
Satisfaction with information provided by health professionals (1–5)	353	−1.39 (0.45)	−2.28–0.50	**0.002**	−1.41 (0.45)	−2.30–0.52	**0.002**
Satisfaction with empathy showed by professionals when informed (1–5)	353	−2.01 (0.43)	−2.87–1.15	**<0.001**	−2.02 (0.43)	−2.88–1.16	**<0.001**

Bold: p≤0.05. Adjusted model: Analyses were adjusted for medical positive COVID-19 test and age. β, unstandardized coefficient. ^a^ Due COVID-19.

## Data Availability

The data presented in this study are available on request from the corresponding author.

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
