# Peer review of "Anxiety and Worries among Pregnant Women during the COVID-19 Pandemic: A Multilevel Analysis"

_ijerph, 2021, doi:10.3390/ijerph18136875_

Round 1

Reviewer 1 Report

The manuscript “Anxiety and worries among pregnant women during the COVID-19 pandemic: A multilevel analysis” assess the associations between socioeconomic and demographic factors, and health concerns and health information management, with anxiety during the COVID-19 pandemic in pregnant Spanish women. The manuscript covers important issues. However, some changes should be done before publication:

  1. Introduction

The introduction should be more focused on the specific problem under study. Thus, sociodemographic factors and health concerns and health information are presented very briefly, while the perinatal period, that “is defined as the period beginning at 22 completed weeks of gestation and ending seven completed days after birth” is presented in greater depth. But the study sample comprises women in the first, second and third trimesters of pregnancy.

  1. Method

2.2. Measurement Instruments

2.2.1. Anxiety level

The reason why anxiety has been assessed with the STAI state (STAI-S) should be explained.

2.2.5. Health information-related factors

This subsection is confusing and all the response options should be specified. It should also be specified how the information was coded and how it was treated when women obtained the information from more than one source.

  1. Results

Table 1 (pages 5-7) is very confusing. Table should be reviewed and made clearer.

Table 2 (pages 8-9) contains two models: Model 0 and Model 1 but there is no information, in the table neither in the text nor in the subsection data analysis, of Model 0. Model 0 should be explained.

All results should be thoroughly reviewed as some errors have been detected. Thus, on page 7 it says: “The STAI-S score was higher in women who have reduced working hours (27.8+/-10.9, p=0.005) and income (23.7+/-10.1, p=0.025) due to the COVID-19…. " But he data relating to income do not coincide with those of Table 1, since it shows “Reduced income” “Yes” “26.4(10.8)”. And in the first line of page 8 it says “(β = 2.608 0.59(SE)” for Delivery and in Table 2 the β value that appears is “2.08”.

References

The References section should be reviewed as some references have been found to be incomplete. For example, on page 13 the reference 50 and 51. And on page 14 the 57 and 59.

Author Response

Thanks for giving us the possibility to re-submit to International Journal of Environmental Research and Public Health a revised version of the manuscript entitled "Anxiety and worries among pregnant women during the COVID-19 pandemic: A multilevel analysis.” by Esteban-Gonzalo et al. (ijerph-1252407).

The authors thank the reviewers for the thoughtful comments and suggestions on our manuscript. We have been able to incorporate all of the feedback in our revisions and feel that the paper is stronger as a result. 

All the authors have approved the publication of the paper in its present form.

We hope that you will find our work of interest.

Sincerely,

ANSWER TO THE REVIEWER’S COMMENTS

Reviewer(s)' Comments to Author:

Referee: 1

The manuscript “Anxiety and worries among pregnant women during the COVID-19 pandemic: A multilevel analysis” assess the associations between socioeconomic and demographic factors, and health concerns and health information management, with anxiety during the COVID-19 pandemic in pregnant Spanish women. The manuscript covers important issues.

Thanks for your words and the general support for our study. We have taken into consideration your comments throughout the manuscript in all the sections to improve. Please, find below the changes for each of the comments.

However, some changes should be done before publication:

  1. Introduction

The introduction should be more focused on the specific problem under study. Thus, sociodemographic factors and health concerns and health information are presented very briefly, while the perinatal period, that “is defined as the period beginning at 22 completed weeks of gestation and ending seven completed days after birth” is presented in greater depth. But the study sample comprises women in the first, second and third trimesters of pregnancy.

Thank you for this recommendation. Extra information regarding sociodemographic factors and health concerns has been added in the introduction. Lines 108-129.

Additionally, the authors appreciate the observation regarding the perinatal period, and we have decided to remove this information as it can generate confusion among the readers, focusing more just on concepts such as pregnancy and postpartum mental illnesses.

  1. Method

2.2. Measurement Instruments

2.2.1. Anxiety level

The reason why anxiety has been assessed with the STAI state (STAI-S) should be explained.

Thank you very much for this comment. The State Trait Anxiety Inventory (STAI) was used to assess anxiety level in participants. This tool was employed due to three different reasons:

  • First, it has been commonly used in relevant studies, widely cited in the literature and adapted for use in a large number of countries.
  • Second, it is a well-documented scale able to diagnose anxiety in clinical settings and can be used to detect distress.
  • Third, the Spanish adaptation has shown adequate psychometric properties, with a Cronbach's alpha reliability of 0.90 for Trait and 0.94 for State Anxiety, and has also been sensitive to increased environmental stimuli that produce stress.

Further, in the present study only STAI state (STAI-S) was used due to limitations in the length of the questionnaire.

This information has been included in the method section. Lines 186-192.

2.2.5. Health information-related factors

This subsection is confusing and all the response options should be specified. It should also be specified how the information was coded and how it was treated when women obtained the information from more than one source.

Thank you to the reviewer for this relevant suggestion, the information provided was not clear enough. We have included extra information in order to clarify this section. Lines 231-257.

  1. Results

Table 1 (pages 5-7) is very confusing. Table should be reviewed and made clearer.

Thank you for the comment, the table has been revised to give greater clarity to the results shown.

Table 2 (pages 8-9) contains two models: Model 0 and Model 1 but there is no information, in the table neither in the text nor in the subsection data analysis, of Model 0. Model 0 should be explained.

Authors agree with the reviewer, the two models built were not well explained. With the objective of solving this problem, we have renamed the models (unadjusted and adjusted models) and included extra information in the methods section (Data analyses). Lines 285-293.

In addition, we have included and explanation in the results section where the results are presented, in order to clarify that, since no relevant differences were detected between the unadjusted model and the adjusted model, only the results associated with the adjusted model will be shown. Lines 323-326.

All results should be thoroughly reviewed as some errors have been detected. Thus, on page 7 it says: “The STAI-S score was higher in women who have reduced working hours (27.8+/-10.9, p=0.005) and income (23.7+/-10.1, p=0.025) due to the COVID-19…. " But he data relating to income do not coincide with those of Table 1, since it shows “Reduced income” “Yes” “26.4(10.8)”. And in the first line of page 8 it says “(β = 2.608 0.59(SE)” for Delivery and in Table 2 the β value that appears is “2.08”.

Thank you very much for your comment, we have proceeded to review the section and correct the errors detected.

  1. References

The References section should be reviewed as some references have been found to be incomplete. For example, on page 13 the reference 50 and 51. And on page 14 the 57 and 59.

Thank you. The mistaken references have been revised.

--------Thanks for your comments and suggestions------

Reviewer 2 Report

I think the authors address an important topic in this article, i.e. the mental health, specifically anxiety levels, of pregnant women during the covid-19 pandemic (in Spain). They investigate whether sociodemographic variables and health-information related variables.  I have some concerns with both the presentation of the method & findings, as well as the theoretical & empirical background of the study:

  • The aim of the paper and the research questions are unclear, and in my opinion, do not logically result from the review of the literature. There must be a more specific underlying research question and/or expectations than the generic sentence on page 3 ("For all these reasons, the objective of the present study was to assess the associations between socioeconomic and demographic factors, health concerns and health information management, and anxiety level during the COVID-19 pandemic in pregnant Spanish women"). Please explain, using relevant literature, to the reader why you:
    • a) focus only on anxiety and not on general mental health/psychopathologies.
    • b) include health-information variables in your research design - apparently, you think they are relevant to pregnant women's anxiety levels, but you should explain why (and support this with literature)
    • c) similarly, it is unclear why you expect that sociodemographic variables influence anxiety levels, and 
    • d) which sociodemographics in particular could be relevant - age, socio-economic background, rural/urban living (population density --> more risk for covid infection?) etc. etc.
  • Smaller issues with the introduction:
    • I think the introduction could start on page 2 with the sentence "the public health .." - I feel the paragraph preceding this one is superfluous.
    • Please avoid terminology & statements like 'social and economic drama' (p.2) - describe the situation objectively but do not attribute feelings or evaluations like this.
  • Methods: it took me some time to realize that the Anxiety measure is your (main?) dependent measure. It would help readers if there was a bit more structure in the sense of indicating which are the 'core' variables and which are covariates.
  • Methods: perhaps a Table with an overview of all questions & answer scales/answer options, including (if relevant) cronbach's alpha could be included for the reader? 
  • Results: Given the lack of a a research question/research aim and/or specific expectations, the results section also are very exploratory. It would really help the reader if the results could be presented using e.g. subheadings in which the effect of the various variables on anxiety are presented in a more structured way.
  • Table 1 is very unclear and could perhaps be split into multiple tables. For example, there are significance levels/tests mentioned for "COVID-19 symptoms and complications","Contagion and consequences on the baby" etc. --> the footnote indicates that these significance results refer to correlation tests. But which variables are these questions correlated with? This remains completely unclear. 
  • Results: please report the findings in the text OR the Tables - not both. The results section seems quite chaotic and given the multitude of results, needs some structuring to help readers identify the most relevant variables.
  • Did you correct for repeated/multiple testing? E.g. Bonferroni correction or reducing the significance level to .01? Or were all variables inserted into 1 large analysis?
  • Was there any attempt to reduce the number of predictors, e.g. by creating average scores for 'covid-19 specific worries', 'healthcare interactions & communications' etc? It is easy to get lost in Table 2 at the moment because the list of predictors is so enormous.
  • Discussion: please repeat the aim(s) of the study before summarizing the results.
  • Generally, the authors seem to equate their finding on the STAI with 'anxiety about covid-19' - but as far as I understood they did not alter the wording of the STAI to reflect covid-19 anxiety. Since mediation analyses are not reported, I would refrain from drawing those kinds of conclusions. Base-line anxiety levels and individual differences in anixety could have also played a role. 
  • Discussion could include a section on practical implications - how can counselors but also obstetric care providers use the information/results from this study to help their pregnant clients? 

Author Response

Thanks for giving us the possibility to re-submit to International Journal of Environmental Research and Public Health a revised version of the manuscript entitled "Anxiety and worries among pregnant women during the COVID-19 pandemic: A multilevel analysis.” by Esteban-Gonzalo et al. (ijerph-1252407).

The authors thank the reviewers for the thoughtful comments and suggestions on our manuscript. We have been able to incorporate all of the feedback in our revisions and feel that the paper is stronger as a result. 

All the authors have approved the publication of the paper in its present form.

We hope that you will find our work of interest.

Sincerely,

ANSWER TO THE REVIEWER’S COMMENTS

Reviewer(s)' Comments to Author:

Referee: 2

Comments to the Author

I think the authors address an important topic in this article, i.e. the mental health, specifically anxiety levels, of pregnant women during the covid-19 pandemic (in Spain). They investigate whether sociodemographic variables and health-information related variables. 

Thank you very much for your words and the general support for our study. We have carefully considered your comments and suggestions in order to make the manuscript stronger. Please, find below the changes for each of the comments.

I have some concerns with both the presentation of the method & findings, as well as the theoretical & empirical background of the study:

  1. The aim of the paper and the research questions are unclear, and in my opinion, do not logically result from the review of the literature. There must be a more specific underlying research question and/or expectations than the generic sentence on page 3 ("For all these reasons, the objective of the present study was to assess the associations between socioeconomic and demographic factors, health concerns and health information management, and anxiety level during the COVID-19 pandemic in pregnant Spanish women").

Thank you very much for this important comment. As suggested, the research question and objective have been remarkable at the end of the introduction. Lines 149-155.

  1. Please explain, using relevant literature, to the reader why you:
  2. a) focus only on anxiety and not on general mental health/psychopathologies.

Thank you for this important question. We decided to choose the STAI state scale because it assesses the existence of threat-related stimuli in the environment of the respondent around the time of the assessment. It works as an equivalent to a measure of negative affect or a stress measure, considering the amount of stressful stimulus, related with several psychological disorders. Therefore, we decided that it was a good idea to use the STAI state scale in order to capture impact of the moment we were living and its effects in vulnerable population such as pregnant women are. In addition, we considered that focusing on anxiety was a good idea because, at the moment we started to collect the data, just few research works were based in analyzing anxiety.

In any case, extra information regarding the test and its properties has been added in the method section. Lines 186-192.

  1. b) include health-information variables in your research design - apparently, you think they are relevant to pregnant women's anxiety levels, but you should explain why (and support this with literature)

Thank you for this recommendation. Extra information regarding health concerns and health information has been added in the introduction. Lines 119-128.

  1. c) similarly, it is unclear why you expect that sociodemographic variables influence anxiety levels, and d) which sociodemographics in particular could be relevant - age, socio-economic background, rural/urban living (population density --> more risk for covid infection?) etc. etc.

Thank you for this recommendation. Extra information regarding sociodemographic factors has been added in the introduction. Lines 108-118.

Smaller issues with the introduction:

I think the introduction could start on page 2 with the sentence "the public health .." - I feel the paragraph preceding this one is superfluous.

Thank you. The first paragraph of the introduction has been omitted.

Please avoid terminology & statements like 'social and economic drama' (p.2) - describe the situation objectively but do not attribute feelings or evaluations like this.

Authors agree with the reviewer, this unfortunate expression has been revised. Line 52.

  1. Methods: it took me some time to realize that the Anxiety measure is your (main?) dependent measure. It would help readers if there was a bit more structure in the sense of indicating which are the 'core' variables and which are covariates.

Thank you for this relevant suggestion. We have included extra information in order to clarify this relevant issue when describing each measurement instrument. Lines 185, 197, 221, 258, 260.

Perhaps a Table with an overview of all questions & answer scales/answer options, including (if relevant) cronbach's alpha could be included for the reader?

Thank you very much for your comment. The authors agree that this table would be of interest. However, due to the large number of scales used in the present work, the table would be several pages long. We believe that this would perhaps be too long. However, if the reviewer considers that this information is nevertheless a priority, we will be happy to construct the table and reflect it in the paper.

  1. Results

Given the lack of a research question/research aim and/or specific expectations, the results section also are very exploratory. It would really help the reader if the results could be presented using e.g. subheadings in which the effect of the various variables on anxiety are presented in a more structured way.

Thank you for the suggestion. The whole results section has been revised in order to provide greater clarity. It has been structured based on subheadings coinciding with the different groups of variables studied and reflected in the objective. We hope that this new structure will be easier for the reader.

Table 1 is very unclear and could perhaps be split into multiple tables. For example, there are significance levels/tests mentioned for "COVID-19 symptoms and complications","Contagion and consequences on the baby" etc. --> the footnote indicates that these significance results refer to correlation tests. But which variables are these questions correlated with? This remains completely unclear.

Thank you for the comment. Table 1 together with its tittle has been edited in order to clarify the understanding of the blocks of the independent variables examined. We hope these changes will be useful.

Results: please report the findings in the text OR the Tables - not both. The results section seems quite chaotic and given the multitude of results, needs some structuring to help readers identify the most relevant variables.

Thank you very much for your comment. The results shown in the text are those that are statistically significant only. We have tried to eliminate the information associated with the bivariate analysis from the section, but it seems strange to omit them altogether. Partial elimination is controversial since we cannot mention some of the statistically significant associations detected and not others. Finally, we can eliminate all the numerical data presented for both bivariate and multivariate analysis, but we found it strange on review to read the results without supporting significance data. Therefore, after several attempts, we have respected the information presented from the beginning. However, if the reviewer has other criteria, authors will be pleased to proceed as you instruct us to do so. Nevertheless, we hope that the new structure of the section will facilitate understanding to some extent.

Did you correct for repeated/multiple testing? E.g. Bonferroni correction or reducing the significance level to .01? Or were all variables inserted into 1 large analysis?

Thank you very much for your questions. Following the indications of our statisticians, instead of performing Bonferroni tests to compare two by two each category studied in those qualitative variables with more than two categories, we chose to perform a multilevel regression. This strategy offered us predictive power and the possibility of acting on the levels of association present in the sample (hospitals). To this end, we ran several different models, one for each of the independent variables studied, in relation to the dependent variable (anxiety levels). In the case of qualitative independent variables with more than two categories, dummy variables were constructed in order to confront each category against all the others. Therefore, in this case, separate regression models were performed for each category of the variable. Regarding the significance level, out statisticians considered correct to set it at 0.05, since assuming an error of 5% is acceptable in health and social science research.

Was there any attempt to reduce the number of predictors, e.g. by creating average scores for 'covid-19 specific worries', 'healthcare interactions & communications' etc? It is easy to get lost in Table 2 at the moment because the list of predictors is so enormous.

Thank you for your comment. Table 2 has been edited in order to clarify the understanding of the blocks of the independent variables examined.

As the reviewer indicates, the number of predictor variables is very high. We understand that this has the disadvantage that it can be overwhelming to review so many variables. When we started working on this study, we did indeed consider reducing the variables studied. However, due to the particular situation that the pregnant women were living, we decided to take on the problem of assessing many variables, in order to have the possibility of knowing in a more concrete way those factors that could influence the anxiety levels of the pregnant women. We considered that all the variables and categories studied provided differentiated information that was of interest. However, if the reviewer has other considerations, we will be happy to take them into account for this and/or future studies. 

  1. Discussion: please repeat the aim(s) of the study before summarizing the results.

The aim of the study has been referred before summarizing the results, as indicated. Thank you.

Lines 357-360.

Generally, the authors seem to equate their finding on the STAI with 'anxiety about covid-19' - but as far as I understood they did not alter the wording of the STAI to reflect covid-19 anxiety. Since mediation analyses are not reported, I would refrain from drawing those kinds of conclusions. Base-line anxiety levels and individual differences in anixety could have also played a role.

The authors appreciate this observation. We would have liked to include measures of anxiety that were specific to the covid-19 period, however, at the moment that we collected the data, no specific measures were available. Therefore, we decided to use the STAI for its excellent properties to capture and discriminate anxiety states. In any case, this has been included as a limitation.  Lines 447-449.

Discussion could include a section on practical implications - how can counselors but also obstetric care providers use the information/results from this study to help their pregnant clients?

Thank you for this interesting contribution, the information indicated has been added to the discussion section, immediately after the conclusions. Lines 468-475.

--------Thanks for your comments and suggestions------

Round 2

Reviewer 1 Report

The revised manuscript "Anxiety and worries among pregnant women during the COVID-19 pandemic: A multilevel analysis" has improved significantly over the first version. Therefore, I think it can be published.

Author Response

ANSWER TO THE REVIEWER’S COMMENTS

Reviewer(s)' Comments to Author:

Referee: 1

The revised manuscript "Anxiety and worries among pregnant women during the COVID-19 pandemic: A multilevel analysis" has improved significantly over the first version. Therefore, I think it can be published.

Thank you very much for the general support for our study. The manuscript is stronger, as a result.

--------Thanks for your comments and suggestions------

Reviewer 2 Report

Thank you for revising your manuscript so thoroughly, I think it has improved considerably. I still have questions about your analyses, however - mainly, which 'levels' did you identify in your multilevel linear regression? How can I identify, from the text or the table(s), these levels? Also, why is a multilevel approach the correct way to analyze these data? Are some of your variables nested? If so, which one(s)? 

Author Response

ANSWER TO THE REVIEWER’S COMMENTS

Referee: 2

 Comments to the Author

Thank you for revising your manuscript so thoroughly, I think it has improved considerably.

Thanks for your words and the general support for our study. We have taken into consideration your comments. Please, find below the changes for each of the comments.

I still have questions about your analyses, however - mainly, which 'levels' did you identify in your multilevel linear regression?

How can I identify, from the text or the table(s), these levels?

Also, why is a multilevel approach the correct way to analyze these data? Are some of your variables nested? If so, which one(s)?

Thank you for your comments. The level whose random intercept was included in the models built for the present study was the hospital where the participant was recruited (Quirónsalud University Hospital of Madrid, Jiménez Díaz Foundation University Hospital and San José Quirónsalud Hospital).

The information regarding the level of association considered was referred to in the data analyses, as part of the methods section. Lines 284-285  (“All models included a random intercept for the hospital where the participant was recruited.”). However, if the reviewer considers that the information should also be referred to in the tables or other sections of the paper, we will be pleased to provide this information.

Authors consider multi-level, mixed-effects linear regression as a very useful statistical tool, which allows us to address the association that could exist between the pregnant women users of each hospital.

There were no nested variables considered in the analyses.

--------Thanks for your comments and suggestions------
